# Athletes’ Mesenchymal Stem Cells Could Be the Best Choice for Cell Therapy in Omicron-Infected Patients

**DOI:** 10.3390/cells11121926

**Published:** 2022-06-14

**Authors:** Mona Saheli, Kayvan Khoramipour, Massoud Vosough, Abbas Piryaei, Masoud Rahmati, Katsuhiko Suzuki

**Affiliations:** 1Department of Anatomical Sciences, and Pathology and Stem Cell Research Centre, Afzalipour Faculty of Medicine, Kerman University of Medical Sciences, Kerman 7616914115, Iran; m.saheli@kmu.ac.ir; 2Neuroscience Research Center, Institute of Neuropharmacology, and Department of Physiology and Pharmacology, Afzalipour School of Medicine, Kerman University of Medical Sciences, Kerman 7616914115, Iran; 3Student Research Committee, Kerman University of Medical Sciences, Kerman 7619813159, Iran; 4Department of Regenerative Medicine, Cell Science Research Center, Royan Institute for Stem Cell Biology and Technology, ACECR, Tehran 1665659911, Iran; masvos@royaninstitute.org; 5Experimental Cancer Medicine, Institution for Laboratory Medicine, Karolinska Institute, 17177 Stockholm, Sweden; 6Department of Biology and Anatomical Sciences, School of Medicine, Shahid Beheshti University of Medical Sciences, Tehran 1985717443, Iran; piryae@sbmu.ac.ir; 7Department of Tissue Engineering and Applied Cell Sciences, School of Advanced Technologies in Medicine, Shahid Beheshti University of Medical Sciences, Tehran 1434875451, Iran; 8Department of Physical Education and Sport Sciences, Faculty of Literature and Human Sciences, Lorestan University, Khoramabad 6815144316, Iran; rahmati.mas@lu.ac.ir; 9Faculty of Sport Sciences, Waseda University, Tokorozawa 359-1192, Saitama, Japan

**Keywords:** stem cell, sport, COVID-19, exercise, omicron

## Abstract

New severe acute respiratory syndrome coronavirus 2 (SARS-CoV-2) variant, Omicron, contains 32 mutations that have caused a high incidence of breakthrough infections or re-infections. These mutations have reduced vaccine protection against Omicron and other new emerging variants. This highlights the need to find effective treatment, which is suggested to be stem cell-based therapy. Stem cells could support respiratory epithelial cells and they could restore alveolar bioenergetics. In addition, they can increase the secretion of immunomodulatory cytokines. However, after transplantation, cell survival and growth rate are low because of an inappropriate microenvironment, and stem cells face ischemia, inflammation, and oxidative stress in the transplantation niche which reduces the cells’ survival and growth. Exercise-training can upregulate antioxidant, anti-inflammatory, and anti-apoptotic defense mechanisms and increase growth signaling, thereby improving transplanted cells’ survival and growth. Hence, using athletes’ stem cells may increase stem-cell therapy outcomes in Omicron-affected patients.

## 1. Introduction

Severe acute respiratory syndrome coronavirus 2 (SARS-CoV-2) has been transmitting worldwide and has become a global pandemic [1]. In addition to pneumonia, SARS-CoV-2 has the potential to harm other organs such as the heart, liver, and kidneys, as well as the blood and immune systems, which may lead to multiorgan failure and death [1,2]. The emerging new variants have worsened this situation because they are more resistant to the vaccination and have high transmissibility. Discovering a new variant (i.e., B.1.1.529/BA.1) on 24 November 2021, caused widespread panic worldwide [3].

The World Health Organization (WHO) Technical Advisory Group on SARS-CoV-2 Virus Evolution classified B.1.1.529 as the fifth variation of concern (VOC) and called it Omicron, on November 26th [4]. The primary viral component that affects the virus’s infectivity and antigenicity, the spike protein, contains 32 mutations in this variant [4]. The intricate mutations in the spike have been postulated as a possible reason for immunity escape and a high incidence of Omicron breakthrough infections or re-infections [4]. As a result, researchers suggest that vaccine protection against Omicron or the new emerging variants may be reduced [5]. Despite emerging newer variants such as XE, the number of Omicron infection per day is by far the highest among all variants [6]. The very high rate of transmissibility and mortality as well as its vaccine-resistant properties have made Omicron the first priority in the COVID19 studies. After uncovering the main mutations in Omicron, as described earlier, researchers have suggested that stem-cell therapy could be one of the most effective treatments [7,8]. However, the effectiveness of stem-cell therapy is limited by the low survival rate of transplanted cells [9]. We believe that exercise training could positively affect stem cells in the athlete’s body, increasing their survival rates after transplantation. Therefore, we suggest using athletes’ stem cells in treating Omicron-affected patients. We will explain the rationale behind this suggestion in this article.

## 2. Cell Therapy as a New Approach to Treatment of Omicron-Infected Patients

According to previous studies, cell-based treatment may be a promising therapeutic option for lung injury such as acute respiratory distress syndrome (ARDS) [10,11]. Stem cells can self-renew and differentiate into multiple cell types, making them an appealing option for cell therapy. Many studies have promoted stem-cell therapy as one of the emerging treatments for refractory diseases with no recognized treatments, including viral infections such as COVID-19 [12,13]. Particular attention has been placed on mesenchymal stem cells (MSCs) because of the ethical and legal limitations associated with other stem cells [14,15]. The majority of registered stem-cell therapy clinical trials have proposed using MSCs as a treatment modality for COVID-19 patients [16,17,18]. Effects of MSC therapy in the lung are associated with the secretion of anti-microbial peptides and proteins, immunomodulatory and antiapoptotic cytokines, and several growth factors and extracellular vesicles [19,20,21]. In addition, the protective effects of MSCs also include direct cell–cell interaction with respiratory epithelial cells, restoring alveolar bioenergetics [22]. Furthermore, keratinocyte growth factor and angiopoietin-1 secreted by MSCs have been demonstrated to repair alveolar-capillary walls in ARDS caused by viral infection [23,24,25]. Leng et al. [26] reported that MSC transplantation reduced levels of C-reactive protein 10-fold, increased oxygen saturation by 89–98% with a reduction in fever, and reduced shortness of breath and pneumonia infiltration, with COVID-19 patients testing negative on the 13th day after MSC transplantation and the immune profile improved with a decrease in pro-inflammatory and an increase anti-inflammatory cytokines. Furthermore, there was an absence of angiotensin-converting enzyme 2 (ACE2) and transmembrane serine protease 2 (TMPRSS2), with high expression of certain trophic factors suggested as other possible immunomodulatory mechanisms of MSCs. Added to this, Orleans et al. [27] and Liang et al. [28] considered immunomodulatory properties of MSCs as the primary mechanism of action in COVID-19 patients with no known adverse or hypersensitivity reactions.

## 3. The Factors That Make Cell Therapy Less Effective

Experimental and clinical studies have demonstrated that the main obstacles in cell therapy results are low cell survival and low cell growth rate [29,30]. After transplantation, cells are exposed to a hostile and inappropriate microenvironment that includes ischemia (a lack of oxygen and nutrients), inflammation, and oxidative stress (superoxide anions and hydrogen peroxide) [31].

Evidence has shown that MSCs have become apoptotic following transplantation and that fewer than 1% of transplanted cells survive four days after injection. Several methods, including cell preconditioning and tissue engineering, have been used to increase transplanted cells’ survival [32]. Cellular preconditioning has been suggested as the most effective method [33,34]. Different preconditioning methods, such as sub-lethal exposure to hypoxia, heat shock, and cytokines have been used to strengthen MSCs [35,36]. Exposure to a stressful environment can make the cells more resistant, helping them to survive in an inappropriate niche after transplantation [37]. Cell survival can also be enhanced by application of a supportive scaffold to deliver therapeutic cells safely to the wound site and to shield the delivered cells from immune system attack while retaining permeability to therapeutic, signaling, and metabolic factors. Scaffolds facilitate connection between hostile vasculature and promote cells survival. However, studies have shown that using the preconditioning method could enhance regenerative function of stem cells by controlling the fate and function of stem cells [32,38].

## 4. Exercise Training Could Increase Stem-Cell Therapy Outcomes

Exercise training is considered a helpful preconditioning method in which MSC donors participate in a structured, purposeful exercise training program, or, MSC donors are themselves athletes [39,40,41]. It has been shown that four weeks of low- and high-intensity exercise training can increase the number of stem cells and increase newly formed cardiomyocytes by 4% to 7% [42,43]. Exercise training could upregulate specific growth factors and cytokines such as the insulin-like growth factor-1 (IGF-1) and the transforming growth factor-beta1 (TGF-β1), neuregulin-1 (NRG-1), periostin (POSTN), and platelet-derived growth factor (PDGF), and their associated signaling pathways in cardiac stem cells [42]. Among these, the IGF-1-phosphoinositide 3-kinase (PI3K)/serine/threonine kinase Akt (protein kinase B) signaling pathway is considered the most well-known cascade, and may explain the positive effect of exercise on MSC therapy, not only in the heart but also in other tissues, because it can regulate several cellular processes including metabolism, apoptosis, autophagy, aging, and growth, which all contribute to increasing MSC survival and growth [44,45].

Exercise training can play an important role in the function and fate of stem cells [39]. First, irisin secretion, induced by exercise, could facilitates MCS homing in the target tissue [40,41]. Furthermore, irisin provides antiapoptotic effect on MSCs by the extracellular-signal-regulated kinase1/2–superoxide dismutase 2 (ERK1/2-SOD2) pathway because ERK1/2 inactivation and SOD2 knockdown abolish the antiapoptotic effect [40].

Second, exercise preconditioning reduces cells apoptosis and promotes cell survival and growth by improving mitochondrial repair and biogenesis, consequently ameliorating cellular respiration [39,46]. Studies have shown that MSC potency for proliferation and differentiation enhances after exercise training [41]. 

Third, exercise training can increase the expression of proangiogenic factors thereby promoting angiogenesis in MCSs.

Fourth, paracrine function of MSCs is reinforced with exercise [40]. Strong evidence suggests that higher levels of physical activity elicit anti-inflammatory effects, thereby lowering the levels of inflammatory cytokines such as IL-6, TNF-α, IL-1β, and CRP [47,48,49,50,51], which, in turn, increase MSCs survival.

Fifth, exercise training is shown to increase nitric oxide (NO) production by overexpression of nitric oxidase synthase. NO has anti-apoptotic effects, and can increase MSC paracrine activity and MSC immunomodulatory properties, thereby improving MSC survival [45].

Sixth, studies have shown that regular light to moderate intensity exercise training could gradually strengthen endogenic antioxidant defense mechanisms, diminishing oxidative stress [52,53,54,55]. When reactive oxygen species (ROS) production exceeds cellular antioxidant capability, oxidative stress could trigger apoptosis [56] It is well documented that high levels of physical activity are associated with reduced accumulation and production of ROS and decreased levels of apoptosis [57,58]. In addition, exercise decreases the expression of caspase-3 and -7 and inhibits apoptosis in stem cells [59].

Seventh, c-Kit, a type III receptor tyrosine kinase (RTK), is involved in multiple intracellular signaling. It is mainly considered a stem-cell receptor that participates in vital functions of the mammalian (including human) bodies [60]. c-kit activation plays a critical role in the differentiation, proliferation, and survival of stem cells [46,61,62]. Recent data illustrated that exercise training increases c-Kit expression in stem cells [45,63].

It should be noted that the adaptations induced by exercise training could maintain/increase until 2–6 after weeks detraining [61,64]. In addition, as mentioned earlier, MSCs can be obtained from bone marrow, muscle, fat, brain, and skin [65]. While fat tissue has been used more than other tissues, as it is the most abundant [65], we suggest, instead, using bone marrow from athletes because of the low body fat. We should also keep in mind that cells obtained from younger donors are less susceptible to oxidative damage, age considerably more slowly in culture, and have a higher proliferation rate [62]. Therefore, MSCs obtaining from the bone marrow of young athletes would be the best choice for cell therapy in Omicron-infected patients.

## 5. Conclusions

Current evidence shows that several mechanisms, including ischemia, inflammation, oxidative stress, and cell apoptosis contribute to a low cell-survival rate in MSC therapy. Given the modulatory effects of exercise training on MSCs (e.g., potential anti-inflammatory effects, strengthening endogenic antioxidant defense, and protective effects against apoptosis) in addition to more advantages associated with using bone marrow from young people, using young athletes’ bone marrow could be the best choice for cell therapy in Omicron-infected patients.

## Data Availability

The data presented in this study are openly available in Kerman University of Medical Sciences at https://kmu.ac.ir/en.

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
