# Peer review of "Athletes’ Mesenchymal Stem Cells Could Be the Best Choice for Cell Therapy in Omicron-Infected Patients"

_cells, 2022, doi:10.3390/cells11121926_

Round 1

Reviewer 1 Report

The Opinion article submitted by Saheli et al. titled “Athlete’s mesenchymal stem cell is the best choice for cell therapy in Omicron infected patients” presents the team’s thoughts on the most effective cell therapy treatment option for the Omicron virus. They postulate that isolating cells from young athletes may provide more effective therapeutic outcomes for reasons outlined in their paper.

MSCs are being widely investigated for use as a treatment for COVID-19 infection. The clinical results presented to date, while mixed, tend to suggest that MSCs mediated therapeutic outcomes in infected patients. A logical extension of this data would conclude that MSCs would also be effective against the Omicron variant.

The authors speculate that cells from athletes will have additional benefits to make them more effective against COVID, including regular exercise, which has been shown to increase the number of cardiomyocytes, enhanced cytokine and growth factor production, the elevation of nitric oxide levels, and innate anti-inflammatory effects. While each of these may have scientific validity, they offer no proof that regular exercise does anything to enhance the biology of MSCs. To my knowledge, very few, if any, papers in the literature have addressed this topic. Their thoughts on this matter are pure speculation and not based on scientific data. If the authors provided any evidence that MSCs are impacted by exercise and that it improves MSC-mediated efficacy, then the reviewer could support their opinion.

Reviewer 2 Report

In this manuscript, the authors propose the use of stem cell therapy to restore alveolar bioenergetics through direct cell transfer to airway epithelial cells. The authors summarize the additional advantages of obtaining MSCs from the bone marrow of young athletes, thereby suggesting that the use of bone marrow from young athletes may be the best option for cell therapy in patients with Omicron infection.

Some additional comments are given below:

1.            The last paragraph of the introduction is too simplistic. The authors should introduce the content and significance of their work to readers in this part, so as to make it easier for readers to understand the author's research content.

2.            In the last paragraph of the introduction, there is no need to emphasize two days ago, just introduce the specific time.

3.            In the 3th part, for solutions to deal with low effective cell therapy, the authors are suggested to give more introductions.

Reviewer 3 Report

The authors of this opinion review set out to provide an unified opinion on the importance of athletes' mesenchymal stem cells as the best option for cell 2 therapy in Omicron-infected patients. Covering this issue is of particular importance these days, since it opens the door to additional possible treatments for the SARS-CoV-2 variant. There are only a few minor concerns that need to be fixed before publishing.

1-    Title is very nice, but I’m suggesting some possible adjustments to be more attractive and clearer, for example: A promising stem cell therapy in Omicron infected patients

2-    Abstract is clear and informative; however, I suggest more English editing could be useful here  

3-    The introduction is very nice for scientists which not familiar with the field and summarize the literature very nicely

4-    Cell therapy as the treatment??? for the subtitle is not clear at least for me and should be improved to be more informative on the point you like to explore:  Cell Therapy as a New Approach to the Treatment of-----------

5-    What makes cell therapy less effective and the solution? Too hard to get the point also here. What you want to say???? I guess to use (What makes cell therapy ineffective, and what is the solution?) however still these all subtitle need to be improved and use other question word such as (current limitations and potential strategies for stem cell therapy)

6-    Again (Exercise training as a solution) for what????

7-    In short title and subtitles are very important tools to explore the idea  and attract  the readers to read the article, please give some effort here to better choose good words.  

8-    References are all updated and good.

Round 2

Reviewer 1 Report

The revised commentary is much improved and actually references science related to MSCs.

Reviewer 3 Report

The authors respond clearly and efficiently to all my comments. The opinion could be ready for publication.